# The High-Risk Human Papillomavirus Type Influences the Tissue Microenvironment in Cervical Intraepithelial Neoplasia Grade 2

**DOI:** 10.3390/v15091953

**Published:** 2023-09-19

**Authors:** Mayumi Saito, Aarthi Rajesh, Carrie Innes, Rachael van der Griend, Peter Fitzgerald, Bryony Simcock, Peter Sykes, Merilyn Hibma

**Affiliations:** 1Department of Pathology, Dunedin School of Medicine, University of Otago, Dunedin 9054, New Zealand; mayumi.saito.pc@gmail.com (M.S.); ar889@cornell.edu (A.R.); 2Department of Obstetrics and Gynaecology, University of Otago, Christchurch 8011, New Zealand; carrie.innes@otago.ac.nz (C.I.); bryony.simcock@cdhb.health.nz (B.S.); peter.sykes@otago.ac.nz (P.S.); 3Canterbury Health Laboratories, Christchurch 8011, New Zealand; rachael.vandergriend@cdhb.health.nz; 4Awanui Labs, Dunedin 9054, New Zealand; peter.fitzgerald@awanuilabs.co.nz

**Keywords:** human papillomavirus, tissue microenvironment, high-risk types, cervical precancer

## Abstract

High-risk, cancer-causing human papillomavirus (HPV) types are associated with cervical precancer and cancer. A high proportion of high-risk HPV precancer lesions undergo immune-mediated regression. The purpose of this study was to determine if the tissue microenvironment of HPV16 and 18 (HPV16/18) cervical intraepithelial neoplasia grade 2 lesions differed from other high-risk types (HPV ‘other’). Consistent with other studies, we found that progression to higher-grade disease was more frequent in HPV16/18 lesions when compared with HPV ‘other’ lesions. HPV16/18 lesions were significantly more likely to be indoleamine 2,3,-dioxygenase 1 (IDO1)-positive and were associated with reduced CD8 and FoxP3 T cells in the lesion. In the stroma, reduced Tbet- and CD32-positive cells and increased Blimp1-positive cells were significantly associated with HPV16/18 lesions when compared with HPV ‘other’ types. On analysis of the IDO1-positive tissues, lesional IDO1 was associated with significantly decreased numbers of CD4-, CD8-, and FoxP3-positive cells in the stroma compared with IDO1-negative tissues. These data suggest that IDO1 expression may impair infiltration of CD4, CD8, and FoxP3 cells into the stroma beneath the precancer lesion. Increased expression of IDO1 may contribute to immune avoidance and an increased frequency of disease progression in HPV16- and 18-positive lesions.

## 1. Introduction

Human papillomaviruses (HPVs) are double-stranded DNA viruses that exclusively infect cutaneous and mucosal epithelium. There are over 170 different HPV types, divided into five genera. Types included in the alpha genera are of particular interest because of their association with cancer. Thirteen HPV types within the α5 (type 51), α6 (type 56 and 66), α7 (type 18, 45, 39, and 59), and α9 (type 16, 31, 33, 35, 52, and 58) species are classified by the International Approval and Registration Centre (IARC) as Group 1 carcinogens (sufficient evidence for carcinogenicity in humans) [1].

Neoplastic disease associated with HPV infection has well-defined grades of cervical intraepithelial neoplasia (CIN), from CIN1–3. The high-risk (hr) cancer-causing HPV types are more frequently associated with high-grade disease (CIN2 and 3), and progression is more rapid in HPV16/18-positive tissue, compared to tissue positive with other hr HPV types (HPV31/33/45/52/58) [2]. Nearly half of women with HPV16 that persists for over two years develop precancer in the following five years [3].

Hr HPV types mediate several changes to the keratinocytes that they infect, facilitating S-phase progression and proliferation. Cellular changes include degradation of p53 mediated by the viral E6 protein, and degradation of pRB family members by E7 [4]. In addition, viral immune evasion helps facilitate persistent infection. Several immune evasion mechanisms have been reported, including depletion of Langerhans cells, downregulation of major histocompatibility complex I (MHCI), and disruption of the interferon pathway, including via cGAS-STING [5,6,7]. These mechanisms have been identified in vitro or in mouse models and have generally been associated with HPV16 E6 and/or E7 proteins, and in the case of downregulation of STING, both HPV 16 E5 and E7 [8,9]. The contribution of regulation of the tissue and immune microenvironment to disease associated with other hr HPV types is less clear.

Indoleamine 2,3,-dioxygenase 1 (IDO1) is an immune-suppressive molecule that is upregulated by HPV16 E7 [10]. It is an intracellular cytosolic enzyme that mediates local suppression of T cell proliferation and activation by catalysing tryptophan to kynurenine. Tryptophan degradation and accumulation of kynurenine in the extracellular microenvironment suppresses effector T cell proliferation and activation [11,12]. Inhibition of IDO1 function has been shown to elevate tumour-infiltrating CD8 T cells in the mouse [13], and in vitro [14].

The purpose of this study is to determine if hr HPV type (HPV16/18 vs. hr HPV ‘other’) is associated with differences in the tissue and immune microenvironment in CIN2, including differences in expression of IDO1. We found that HPV16/18 was associated with a significantly higher probability of progression over a 24-month follow-up, whereas HPV ‘other’ disease more frequently regressed. HPV16/18 tissues were also significantly more likely to be positive for IDO1. Additionally, fewer CD8- and FoxP3-positive T cells were present in HPV16/18 lesional tissue and Tbet- and CD32-positive cells were less frequent in the stroma when compared with other hr HPV types. When we looked specifically at the IDO1-positive tissues, the numbers of CD4, CD8, and FoxP3+ cells were reduced compared to IDO1-negative tissues. The marked reduction in the numbers of T cells in the lesional and stromal tissue of HPV16/18 lesions is likely to contribute to less frequent immune-mediated regression, when compared with HPV ‘other’ tissues.

## 2. Materials and Methods

### 2.1. Participants

The patient cohort in this study are a subset of the participants of a larger observational management study, known as the ‘PRINCess’ trial, and have previously been reported [15,16]. For inclusion in the study reported here, women had to meet the following additional criteria: both hrHPV- and p16^INK4A^-positive test results were required and CIN2 had been confirmed in the tissue at review by a second pathologist.

### 2.2. HPV Testing

Participants were separated into HPV16/18-positive or hr HPV ‘other’, based on the result of HPV testing (Abbott Real Time High-risk HPV Assay) of their cytological sample taken prior to biopsy. The Abbott Test detects HPV16, HPV18, HPV ‘other’, and negative. The HPV ‘other’ category includes positivity for one or more of the following 12 HPV types: 31, 33, 35, 39, 45, 51, 52, 56, 58, 59, 66, and 68.

### 2.3. Immunohistochemical Staining

Biopsies of CIN2 lesions obtained from women that met the criteria outlined above were formalin-fixed, paraffin-embedded, and sectioned to 4 µm thick. The antigen retrieval, staining, and evaluation of the sections for the following markers has been previously described: CD4, Tbet, GATA3, interleukin (IL)-17, FoxP3, CD8, granzyme B, langerin, Fascin, CD11c, CD32, CD138, high mobility group box (HMGB)-1, B-lymphocyte-induced maturation protein (Blimp)-1, thymic stromal lymphopoietin (TSLP), IDO1, and programmed death-ligand-1 (PD-L1) [15].

### 2.4. Statistical Analysis

Pairwise comparisons were made using the Mann–Whitney U (M–W U)-test. The chi-square and Fisher’s exact tests were used for categorical measures. Survival curves were plotted using CIN3+ as the endpoint, and the distribution of time to the CIN3+ outcome between the two groups was determined using the log-rank test (Mantel–Cox). Hazard ratios were determined using the Mantel–Haenszel test. Univariate analysis was carried out using simple logistic regression to determine the association between markers and HPV type. A Spearman’s correlation analysis was carried out to test the relationship between specific markers that were tested here. All statistical analyses were carried out using GraphPad Prism (Version 9.0).

## 3. Results

### 3.1. HPV16-Positive Lesions Were Frequently Also Positive for ‘Other’ hr HPV

This study reports results on a subset of 69 HPV-positive participants in the PRINCess study. Thirty of the women in this study were categorised as HPV16/18-positive, and 39 were HPV ‘other’. Of the HPV16/18-positive women, 15 women had only HPV16 (14) or HPV18 (1) and one woman had both HPV16 and HPV18. The remaining 14 HPV16/18 (13 HPV16, 1 HPV18) women had coinfections with other hr HPV types.

The clinical characteristics of the participants separated by HPV16/18 or HPV ’other’ high-risk types are shown in Table 1. The entry criteria for the PRINCess study were women under the age of 25 years old who had diagnosed CIN2, which is reflected in the narrow age range of 17–24 years in both the HPV16/18 and HPV ‘other’ cohorts described here. While smoking rates were higher in the HPV16/18 group compared with the HPV ’other’ group, this difference was not statistically significant. The HPV type (HPV16/18 or ‘other’) that was detected in participants had no significant impact on the reported lesion size.

Expectedly, significantly few vaccinated women were in the HPV16/18 group compared with the HPV ‘other’ group. Vaccination for HPV16 and 18 was introduced in New Zealand in 2008 and the change in genotypes associated with vaccination in the PRINCess study, from which this subset of participants was obtained, has been described in detail elsewhere [17].

While the majority of hrHPV+-vaccinated women were positive for HPV ’other’ rather than HPV16/18 in these cohorts, two vaccinated women were HPV16/18-positive. Of those women, one was positive for HPV16 and another non-HPV18 high-risk type and the other was HPV16- and 18-positive. Both individuals had disease that subsequently became more severe (CIN3+) during follow-up.

There was little apparent evidence for any cross-protection between the HPV16/18 vaccine and the HPV ‘other’ types, as a significantly more substantial proportion of the HPV ‘other’ women were vaccinated and had high-grade disease (CIN2).

### 3.2. HPV16/18-Positive Women Are More Likely to Progress to More Severe Disease than HPV ‘Other’-Positive Women

Participants in this study were CIN2 on entry and were monitored for 24 months. Women were treated and withdrawn from the study if disease became more severe (CIN3+) during that time. The progression to the CIN3+ endpoint for HPV16/18 and HPV ‘other’ women is shown in Figure 1. HPV16/18 women were 4.5 times more likely to progress to the CIN3+ endpoint than women infected with HPV ‘other’ types. This result is consistent with other studies that have identified the significantly greater risk of disease progression if women are infected with HPV16/18 compared with other high-risk types.

### 3.3. E1^E4 Staining Was Infrequent in hrHPV-Positive CIN2 Lesions

The HPV E1^E4 protein is a product of E1 and E4 spliced mRNA and is an abundant protein in the viral lifecycle during the productive phase. HPV16 E1^E4 induces G2/M cell cycle arrest and induces apoptosis in keratinocytes, and is expressed in the upper stratified layers of the epidermis during HPV infection [18,19,20]. While p16 overexpression is a direct result of HPV E7 degradation of pRb, the nuclear expression of Ki67 occurs in the nucleus during cell division as a result of cell proliferation. Ki67 expression is confined to the basal lower third of normal epidermis; however, expression in the upper layers is associated with increasing grades of CIN [21].

We looked at expression of the E1^E4 protein in HPV16/18 and HPV ‘other’ tissues and its association with Ki67 staining (Table 2). Overall, E1^E4 staining (16.9% positivity) was infrequent in the CIN2 lesions. E1^E4 staining was more frequent in HPV ‘other’ compared with HPV16/18 lesions (albeit not significantly; *p* = 0.08; Fisher’s exact test). Of the lesions with Ki67 staining in the lower third only, none of the HPV16/18-positive lesions were positive for E1^E4, compared with 57% of the HPV ‘other’ lesions.

### 3.4. IDO1 Expression Is More Frequent in HPV16/18-Positive Lesions

IDO1 is an enzyme that is overexpressed in many cancer cells and is associated with immunosuppression [22]. PD-L1 is also overexpressed in some types of cancers and is a successfully targeted checkpoint molecule [23]. As both molecules play a role in cancer immune escape, and expression of either of these proteins is generally associated with a poor prognosis in cancer, we wished to determine if expression was increased in HPV16/18 versus HPV ‘other’ infected lesions.

In positive tissues, IDO1 expression in the epidermal cells of the lesion was generally strong, punctate, and perinuclear (Figure 2). There was also weaker diffuse cytoplasmic staining in the majority of the positive cells. Significantly (Fisher’s exact test; *p* < 0.0001), we found that approximately half of the HPV16/18 lesions were positive for IDO1, and that IDO1 expression on HPV ‘other’ tissues was rare (Table 3). In contrast, less than 20% of the tissues were PD-L1-positive, irrespective of hr HPV type.

We also looked at coexpression of PD-L1 and IDO1. While noting that the number of PD-L1-positive tissues was low, all the PD-L1-positive HPV16/18 cases (*n* = 5) were also positive for IDO1, whereas all of the PD-L1-positive cases in HPV ‘other’ (*n* = 7) were IDO1-negative.

### 3.5. Immune Cell Infiltrates Were Reduced in HPV16/18 Lesions Compared with HPV ‘Other’ Lesions

We carried out a detailed analysis of the immune microenvironment of the HPV16/18 and HPV ‘other’ tissues (Table 4). The markers tested were primarily to characterise T cells, B cells, and antigen presenting cells. Additionally, expression of the immune regulatory molecules HMGB1 and Blimp-1 were also assessed.

Significantly, around half the number of CD8 and CD4+FoxP3+ T cells were present in HPV16/18 compared with HPV ‘other’ lesional tissue. Although not statistically significant, for several of the other markers (CD4, granzyme B, Langerin, CD32, and CD138), there were, on average, more positive cells in the HPV ‘other’ compared with HPV16/18 lesional tissue.

When we examined the lamina propria below HPV ‘other’ and HPV16/18 lesions (Table 5), we found that numbers of Tbet- and CD32-positive cells were significantly lower in HPV16/18 compared with HPV ‘other’ lesions. In contrast, Blimp-1-positive cells were significantly more likely to be detected in the lamina propria beneath HPV16/18 lesions when compared with HPV ‘other’ lesions.

### 3.6. The Correlations between Cells Differ in HPV16/18 and HPV ‘Other’ Tissues

We examined the relationships between different cell types in the lesion and the lamina propria beneath the lesion, in particular looking for marked differences between HPV16/18 and HPV ‘other’ tissues (Figure 3). The presence of CD4+FoxP3+ T cells directly correlated most strongly with HMGB1-positive cells in HPV16/18 lesions but more strongly with Blimp-1-positive cells in HPV ‘other’ lesions. Additionally, there was a strong correlation between CD4+ and IL17+ cells, and CD4+ and CD8+ cells, in HPV ‘other’ lesions, which was not apparent in HPV16/18 lesions.

In the lamina propria, Blimp-1-positive cells were inversely correlated with CD32+ cells, irrespective of hrHPV type. CD4-, CD8-, and granzyme B-positive (but not double-positive) cells were correlated with CD11c-positive cells in the HPV ‘other’, but not HPV16/18 lamina propria. Conversely, IL17-positive cells were inversely related with FoxP3+ and CD4+FoxP3+ cells in the HPV16/18 and not HPV ‘other’ lamina propria.

### 3.7. The Lamina Propria Beneath IDO1-Positive Tissue Has Fewer CD4, CD8 T Cells and FoxP3-Positive Cells

We carried out a comparison of the cell populations between IDO1-positive and IDO1-negative lesions in the hr HPV-positive tissues (Table 5). While there was no significant difference in the frequency of particular cell subsets in the lesional tissue, there was a marked reduction in the numbers of cells beneath the lesion. Specifically, the CD4 and CD8 T cells were around half as frequent in the lamina propria beneath the IDO1-negative tissues compared to IDO1-positive tissues. Perhaps surprisingly, the frequency of FoxP3-positive cells in IDO1-positive tissue was around a quarter of that of IDO1-negative tissue. This reduction was also reflected in both the CD4+FoxP3+ and the CD4-FoxP3+ subsets.

## 4. Discussion

In this study, we report the effects of hr HPV type (HPV16/18 and HPV ‘other’) on the immune microenvironment of p16+ CIN2 lesions. We found that HPV16/18 CIN2 lesions, which were more likely to progress, are significantly more frequently IDO1-positive than other hr HPV-positive CIN2 lesions. Furthermore, we found an HPV16/18-associated decrease in lesional CD8+ and CD4+FoxP3+ cells and stromal Tbet+ and CD32+ cells, when compared with other hr HPV types. Additionally, Blimp1 expression in the stroma was increased in the HPV16/18 compared with HPV ‘other’ lesions.

While some HPV infections persist beyond two years, most infections undergo immune-mediated regression. In women with CIN2, others have reported that disease regresses in 50% of cases within two years [24]. In the patient cohort studied here, disease in at least 53% of the women under 25 with CIN2 spontaneously regressed [25]. Importantly, progression was more likely in HPV16/18 infection than with other hr HPV types. A contributing factor may be that the tissue microenvironment of HPV16/18 lesions is not conducive to an effective antiviral immune response.

We only included p16-positive, block-stained lesions in this study. Block staining for p16 (i.e., continuous staining of the cells of the basal and parabasal layers of cervical squamous epithelium, with or without staining in the intermediate or superficial layers) is strongly associated with high-grade hr HPV-positive disease [26]. Increased p16 expression is a result of a pRB-independent increase by E7 of the expression of the histone demethylase KDM6B. Consequential demethylation of the repressive histone mark H3K27me3 occurs, resulting in the induction of p16 [27].

While p16 would typically induce pRB-dependent senescence, E7 degrades pRB, and cells are able to proliferate. Increased proliferating cells, as evidenced by Ki67 staining, was seen in the majority (around 80%) of stained tissues irrespective of HPV type. Considering that only around 15% of the tissues were positive for E1^E4 and that p16 upregulation is a consequence of E7, the viral early promoter, which controls E7, is likely to be more active in these tissues than the late promoter, which drives E1^E4. This is consistent with the predicted cellular changes in viral gene expression that are associated with progression of cells to a more transformed phenotype [28].

We found that IDO1 was significantly more frequently expressed in HPV16/18 lesions compared with hr HPV ‘other’ lesions. In normal cervix IDO1 is expressed only in rare glandular cells but is frequently expressed in cervical carcinomas [29]. Heeren et al. (2018) reported IDO1 expression in primary squamous carcinoma cells and stromal immune cells, as well as in metastatic IDO1+ tumour cells in the lymph nodes [30]. In a study of liquid-based cytology samples, IDO1 was significantly increased in HSIL squamous cells compared with control samples. Consistent with what we observed here, IDO1 staining was detected in the upper layers of the epidermis in CIN [31]. In contrast, in head and neck squamous cell carcinoma, IDO1 expression was most commonly at the focal interface between the tumour and the stroma and was not associated with HPV status (HPV-positive versus -negative) [32].

How IDO1 expression is regulated by HPV16/18 is less clear. Increased IDO1 expression is typically associated with a loss of BIN1 or overexpression of COX2. COX2 expression is increased in cervical cancer [33]. HPV16 E6 and E7 expression induces COX-2 transcription by activating the epidermal growth factor receptor in vitro, and the induction of COX-2 expression may be the mechanism by which HPV16/18 increases IDO1 expression. The infrequent IDO1 expression in other hr HPV lesions suggests that viral regulation of IDO1 may be restricted to only selected hr HPV types.

We found that CD8+ T cells in the lesion and Tbet+ cells in the stroma were reduced in HPV16/18 lesions, compared with hr HPV ‘other’ types. Consistent with these observations, commonly reported high-grade cervical disease associated cellular changes when compared with normal tissue include decreased CD4 and CD8 cells and increased IDO1-positive cells [34]. Changes associated with subsequent disease progression from CIN2 to CIN3 include decreased CD4+, T bet-positive, CD8+, and CD11c+ cells [15]. Overall, the reduced numbers of T cells in HPV16/18 compared with hr HPV ‘other’ tissues in this study may reflect inhibition of T cell proliferation in the local tissue microenvironment. The frequent expression of IDO1 in HPV16/18 tissues may contribute to the reduced numbers of CD8 and CD4+FoxP3+ T cells that proliferated in these lesions. We also found significantly reduced numbers of CD4 and CD8 T cells in the stroma of IDO1-positive tissues, suggesting that the effects on T cell proliferation mediated through IDO1 may extend into the stroma beneath the lesion.

In the skin-expressing HPV16 E7 transgenic mouse model, inhibition of IDO1 actively promoted rejection of E7-skin grafted onto an immune competent mouse that would not otherwise be rejected [10]. This indicates a direct role for HPV16 E7 in the regulation of IDO1, and direct immunosuppression mediated by this molecule. We propose that the more frequent expression of IDO1 in HPV16/18 compared with HPV ‘other’ types in the CIN2 lesions may reduce the frequency of immune-mediated regression in those tissues.

Counterintuitively, we found that higher numbers of CD4+FoxP3+ cells were significantly associated with HPV ‘other’ lesions, whereas HPV16/18 lesions had reduced numbers of these cells. In HPV16/18 lesions, CD4+FoxP3+ cells were strongly correlated with HMGB1, whereas they were strongly correlated with Blimp1 expression in HPV ‘other’ lesions. Previously, we did not find any association between CD4+FoxP3+ cells and disease progression in CIN2+ lesions but did detect an increase in CD4-FoxP3+ cells in lesions that subsequently progressed to higher-grade disease [15]. Furthermore, we found that Blimp1 and HMGB1 expression in the lesion was associated with a higher likelihood of disease progression. Additionally, others have reported that Treg are increased in the stroma in high-grade disease [34]. CD4+FoxP3+ cells demonstrate functional plasticity [35]. We note that CD4+FoxP3+ cells were detected at a low frequency in the lesional tissue (around 10-fold less than the CD8 T cells), and it is unclear what functional impact this comparatively low number of cells may have.

## 5. Conclusions

Cell populations in the tissue microenvironment of CIN2 lesions differ depending on the hr HPV type(s) present (HPV16/18 vs. hr HPV ‘other’). Disease progression is more frequent in HPV16/18-positive lesions and is associated with IDO1 positivity, reduced lesional CD8 cells and lamina propria Tbet- and CD32-positive cells, and increased Blimp1 expression. Surprisingly, CD4+FoxP3-positive cells were also reduced in the lesion and stroma of HPV16/18 CIN2 tissue when compared with HPV ‘other’. We propose that the increased expression of IDO1 and reduction in T and B cells contributes to a reduced likelihood of immune-mediated regression in HPV16/18 tissues. The consequences of the reduction in CD4+FoxP3+-positive cells in HPV16/18 CIN2 stroma are yet to be elucidated.

## Figures and Tables

**Figure 1 viruses-15-01953-f001:**
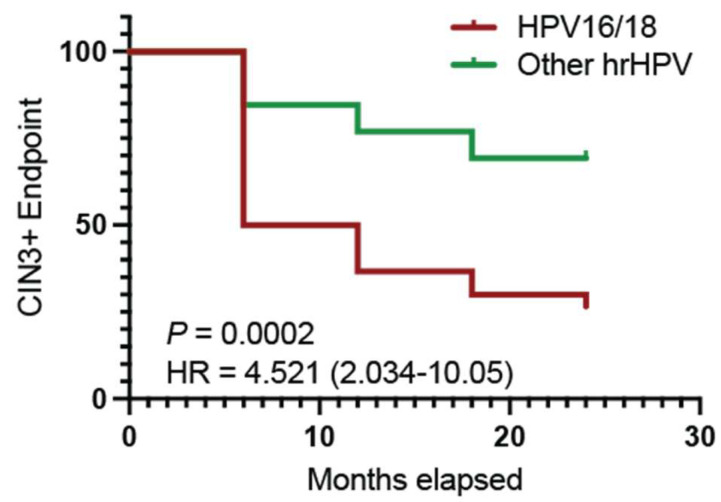
Survival curve with CIN3+ as the endpoint for HPV16/18 and HPV ‘other’.

**Figure 2 viruses-15-01953-f002:**
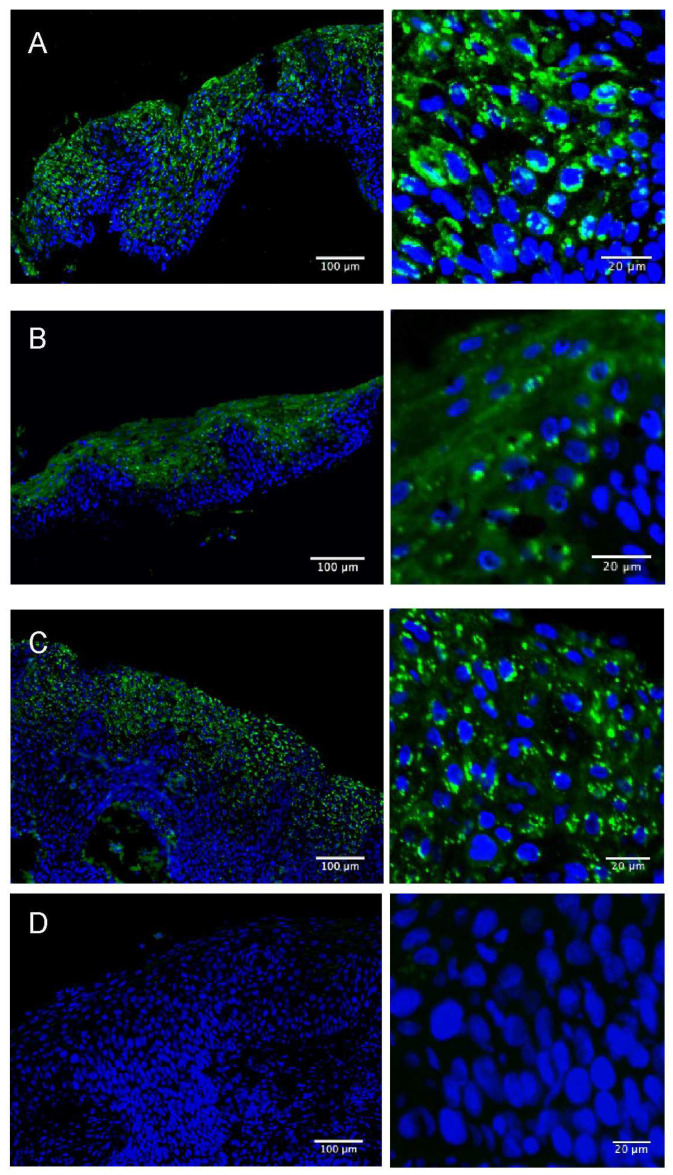
Representative examples of positive (**A**–**C**) and negative (**D**) epidermal IDO1 staining in hr HPV-positive CIN2 tissues.

**Figure 3 viruses-15-01953-f003:**
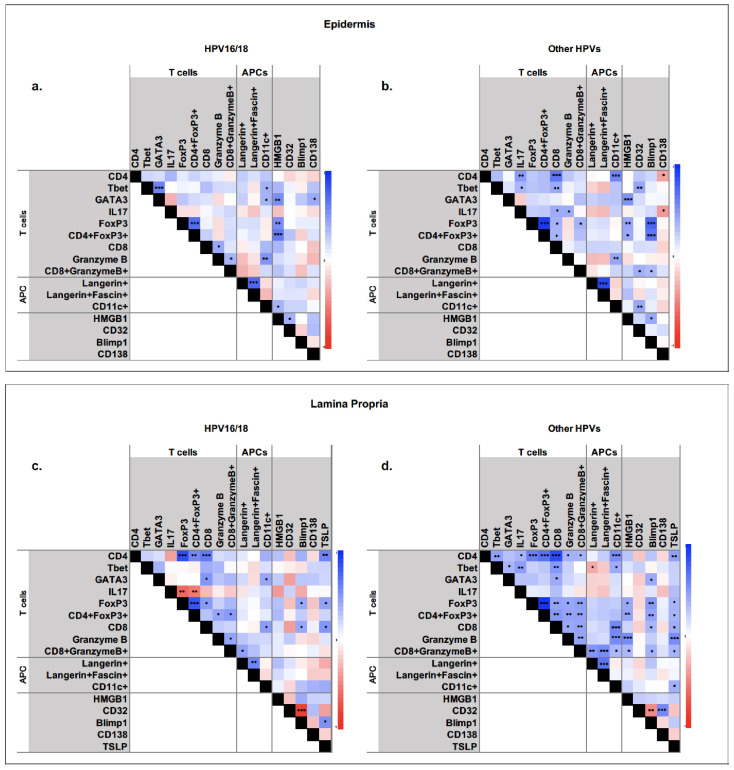
Correlations between variables in hrHPV16/18 and HPV ‘other’ CIN2 lesions (**a**,**b**) and lamina propria beneath the lesions (**c**,**d**). Blue indicates a direct correlation and red indicates an inverse correlation. P values are indicated on each matrix: * *p* < 0.05; ** *p* < 0.01; *** *p* < 0.001.

**Table 1 viruses-15-01953-t001:** Baseline characteristics of HPV16/18 and HPV ’other’ groups.

		HPV16/18 (*n* = 30)	HPV ‘Other’ (*n* = 39)	Statistical Significance
Age	Mean (Range)	21.50 (17–24)	21.91 (19–24)	*p* = 0.1043 (M–W U *)
Smoker	Yes	13 (43.3%)	11 (28.2%)	*p* = 0.4031 (chi-square)
	No	17 (56.7%)	26 (66.7%)	
	Unknown	-	2 (5.1%)	
Vaccine	Yes	2 (6.7%)	25 (64.1%)	*p* < 0.0001 (chi-square)
	No	12 (40.0%)	4 (10.3%)	
	Unknown	16 (53.3%)	10 (25.6%)	
Lesion area	Mean (Range)	0.4334 (0.0093–2.148)	0.3812 (0.0112–1.672)	*p* = 0.9462 (M–W U *)

* Mann–Whitney U.

**Table 2 viruses-15-01953-t002:** E1^E4 and Ki67 expression in hr HPV lesions.

HPV16/18 (*n* = 30)				HPV ‘Other’ (*n* = 39)			
Ki67		E4		Ki67		E4	
**1/3 ***	5 (16.7%)	**E4+**	0 (0%)	**1/3**	8 (21.1%)	**E4+**	4 (57.1%)
		**E4−**	5 (100%)			**E4−**	3 (42.9%)
		**ND ****	1			**ND**	1
**2/3**	23 (76.7%)	**E4+**	3 (13.0%)	**2/3**	24 (63.2%)	**E4+**	4 (17.4%)
		**E4−**	20 (87.0%)			**E4−**	19 (82.6%)
		**ND**	1			**ND**	1
**3/3**	2 (6.7%)	**E4+**	0 (0%)	**3/3**	6 (15.8%)	**E4+**	0 (0%)
		**E4−**	2 (100%)			**E4−**	6 (100%)
		**ND**	-			**ND**	-
**ND**				**ND**	1	**E4−**	1

* Ki67 staining in 1/3 (the lower third), 2/3 (the lower and middle thirds), or 3/3 (full thickness) of the epidermis; ** no data.

**Table 3 viruses-15-01953-t003:** Expression of PD-L1 and IDO1 in HPV16/18 and HPV ‘other’ lesions.

	HPV16/18 (*n* = 30)				HPV ‘Other’ (*n* = 39)			Fisher’s Exact Test
PD-L1								
Positive	5 (16.7%)	* <CIN3	1 (20.0%)		7 (18.4%)	<CIN3	5 (71.4%)	*p* = 0.8505
		CIN3	4 (80.0%)			CIN3	2 (28.6%)	
Negative	25 (83.3%)	<CIN3	7 (28.0%)		31 (81.6%)	<CIN3	21 (67.7%)	
		CIN3	18 (72.0%)			CIN3	10 (32.3%)	
IDO1								
Positive	14 (46.7%)	<CIN3	4 (28.6%)		2 (5.4%)	<CIN3	1 (50.0%)	*p* < 0.0001
		CIN3	10 (71.4%)			CIN3	1 (50.0%)	
Negative	16 (53.3%)	<CIN3	4 (25.0%)		35 (94.6%)	<CIN3	24 (68.6%)	
		CIN3	12 (75.0%)			CIN3	11 (31.4%)	

No data were available for one HPV ‘other’ PD-L1 sample and two HPV ‘other’ IDO1 samples. * <CIN3 indicates persistence or regression at follow-up; CIN3 indicates progression to CIN3 at follow-up.

**Table 4 viruses-15-01953-t004:** Cells in the epidermis and lamina propria of HPV16/18- and HPV ‘other’-positive tissues.

	Epidermis			Lamina Propria		
	Cells/mm^2^ (Mean ± SD)			Cells/mm^2^ (Mean ± SD)		
	HPV16/18	HPV ‘Other’	*p* Value ^#^	HPV16/18	HPV ‘Other’	*p* Value ^#^
CD4	298.13 ± 237.48	407.11 ± 380.94	0.2072	2008.57 ± 1986.88	1993.33 ± 1506.65	0.7521
Tbet	88.74 ± 117.54	151.09 ± 146.81	0.1013	371.78 ± 276.78	834.04 ± 945.68	0.0335 *
GATA3	1579.15 ± 1968.70	1223.91 ± 1346.17	0.7189	3009.15 ± 2279.40	2399.54 ± 1665.51	0.3533
IL17	489.24 ± 664.42	380.62 ± 481.35	0.6273	1280.58 ± 680.08	1613.62 ± 851.25	0.1329
FoxP3	27.23 ± 100.09	24.61 ± 45.77	0.4164	56.27 ± 69.83	79.59 ± 79.14	0.2534
FoxP3+CD4−	21.35 ± 99.16	12.49 ± 39.86	0.8742	17.21 ± 28.94	17.50 ± 24.48	0.6946
FoxP3+CD4+	5.15 ± 9.15	12.12 ± 14.54	0.0151 *	39.06 ± 50.12	62.09 ± 65.94	0.1555
CD8	125.09 ± 104.54	246.16 ± 210.55	0.0086 **	609.07 ± 454.41	888.83 ± 804.86	0.1275
Granzyme B	95.85 ± 127.42	129.85 ± 144.74	0.2477	411.28 ± 417.18	605.03 ± 817.21	0.3997
CD8+ GranzymeB+	0.54 ± 1.27	1.89 ± 4.39	0.1944	11.83 ± 23.91	66.75 ± 227.14	0.1811
Langerin+	21.70 ± 26.78	38.41 ± 58.94	0.194	2.92 ± 5.54	6.05 ± 9.65	0.2664
Langerin+ Fascin+	12.48 ± 14.76	26.91 ± 43.32	0.193	1.10 ± 2.97	2.36 ± 5.65	0.2948
CD11c	87.07 ± 66.39	109.04 ± 126.70	0.8521	545.86 ± 1621.87	353.28 ± 341.68	0.9857
CD32	195.57 ± 473.50	408.62 ± 866.34	0.0644	80.04 ± 74.57	235.32 ± 297.03	0.0052 **
CD138	740.95 ± 642.07	1043.74 ± 1912.31	0.4895	845.84 ± 715.60	1706.27 ± 2549.27	0.1554
HMGB1	1615.30 ± 1559.87	1534.52 ± 1364.22	0.7463	3790.87 ± 1989.52	3841.23 ± 1995.40	0.9285
Blimp1	537.32 ± 996.82	343.93 ± 1094.08	0.1097	1760.00 ± 1563.05	1033.87 ± 1392.86	0.0447 *
TSLP	^a^ ND	ND	ND	1205.08 ± 793.24	941.04 ± 647.56	0.207
Area	0.43 ± 0.49	0.38 ± 0.42	0.748	ND	ND	ND

^a^ ND: not detectable. * *p* < 0.05, ** *p* < 0.01; ^#^ simple logistic regression analysis of square root transformed data.

**Table 5 viruses-15-01953-t005:** Immune cell densities in IDO1-positive and IDO1-negative hr HPV-positive lesions.

	Epidermis			Lamina Propria		
	IDO+ Cells/mm^2^	IDO− Cells/mm^2^	*p* Value ^#^	IDO+ Cells/mm^2^	IDO− Cells/mm^2^	*p* Value ^#^
CD4	304.57	377.51	0.408	1075.66	2265.73	0.01 **
Tbet	92.61	134.41	0.6268	344.55	740.23	0.1176
GATA3	1384.55	1405.66	0.9136	2999.24	2546.18	0.4573
IL17	426.39	425.44	0.7062	1744.54	1377.64	0.1172
FoxP3	10.55	30.84	0.5006	21.80	85.69	0.0006 ***
FoxP3+CD4−	3.53	21.10	0.5685	3.95	22.40	0.0034 **
FoxP3+CD4+	6.79	9.74	0.5781	17.85	63.29	0.0023 **
CD8	163.56	201.05	0.6991	477.46	874.07	0.0304 *
Granzyme B	93.63	117.03	0.6063	357.19	585.70	0.4148
CD8+ GranzymeB+	0.88	1.45	0.7508	10.73	58.24	0.1161
Langerin+	22.00	34.30	0.3968	2.50	5.37	0.4949
Langerin+ Fascin+	14.42	22.83	0.6646	1.29	1.83	0.6469
CD11c	83.50	105.68	0.6951	193.95	506.50	0.084
CD32	195.57	367.30	0.4119	91.10	197.40	0.1652
CD138	616.64	1014.23	0.3613	691.26	1549.87	0.1375
HMGB1	1823.29	1448.84	0.1634	3796.03	3778.15	0.7355
Blimp1	943.97	281.20	0.0796	1634.13	1236.52	0.2363
TSLP	^a^ ND	ND	ND	1105.43	1028.97	0.7995
Area	0.41	0.40	0.5534	ND	ND	ND

^a^ ND: not detectable. * *p* < 0.05, ** *p* < 0.01, *** *p* < 0.001. ^#^ Univariate analysis on square root transformed data.

## Data Availability

The datasets analysed during the current study are available from the corresponding author on reasonable request.

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
