# Peer review of "The High-Risk Human Papillomavirus Type Influences the Tissue Microenvironment in Cervical Intraepithelial Neoplasia Grade 2"

_viruses, 2023, doi:10.3390/v15091953_

Round 1

Reviewer 1 Report

This is a natural history study of CIN2. As observed in other studies they found that CIN2 progression occurred more frequently among women infected with HPV16/18 than with other high risk HPV types. Next, they evaluated the presence of immune cells and markers in the lesions and underlying stroma. IDOT-1 expression was shown to be higher in 16/18 CIN2 than in CIN2 with other HPV types. IDOT-1 expression was inversely correlated with CD8+ and FoxP3 CD4+ cells in lesions. The paper is well written and the data well presented. I only have a few minor suggestions. One is that the figure legend for figure 3 should describe what the red and blue colors indicate (positive/negative correlations). For figure 2 it would be nice to see a negative sample as a control. The majority of women with CIN2 ‘other’ lesions had been vaccinated whereas very few of the women with HPV16/18 infections had been. Although the antibody response to the vaccine is type-specific it would be interesting to know if the authors think that vaccine provided some cellular immunity and protection for women with CIN2 ‘other’?   

Author Response

This is a natural history study of CIN2. As observed in other studies they found that CIN2 progression occurred more frequently among women infected with HPV16/18 than with other high risk HPV types. Next, they evaluated the presence of immune cells and markers in the lesions and underlying stroma. IDOT-1 expression was shown to be higher in 16/18 CIN2 than in CIN2 with other HPV types. IDOT-1 expression was inversely correlated with CD8+ and FoxP3 CD4+ cells in lesions.

The paper is well written and the data well presented. I only have a few minor suggestions.

  • One is that the figure legend for figure 3 should describe what the red and blue colors indicate (positive/negative correlations).

The following text has been added to the figure legend: ‘Blue indicates a direct correlation and red indicates an inverse correlation.’

 For figure 2 it would be nice to see a negative sample as a control.

  • We have added a tissue that was stained with IDO-1 but did not show any positive signal. We considered this to be more informative than a secondary only or isotype control, and hope the reviewer agrees that this is satisfactory.

 The majority of women with CIN2 ‘other’ lesions had been vaccinated whereas very few of the women with HPV16/18 infections had been. Although the antibody response to the vaccine is type-specific it would be interesting to know if the authors think that vaccine provided some cellular immunity and protection for women with CIN2 ‘other’?

  • The reviewer has raised an interesting point. The following statement has been added to the manuscript: ‘There was little apparent evidence for any cross-protection between the HPV16/18 vaccine and the HPV ‘other’ types, as a significantly more substantial proportion of the HPV ‘other’ women that were vaccinated and had high-grade disease (CIN2), compared with the HPV16/18 women.’ 

Reviewer 2 Report

This is a study about the tissue microenvironment in cervical intraepithelial neoplasia grade 2 and high-risk human papillomavirus type. The authors found that increased expression of IDO1 may contribute to immune avoidance and an increased frequency of disease progression in HPV16 and 18 positive lesions.

The paper is well written. However, some issues remain.

All the acronyms must be explained at their first appearance in the text (e.g., IDO1). Moreover, a description of IDO1 should be added in the Introduction.

In table 2, the authors should specify which third (1/3, 2/3, or 3/3) is the lower one.

More Kaplan-Meyer estimator curves according to immune microenvironment may be helpful.

Author Response

This is a study about the tissue microenvironment in cervical intraepithelial neoplasia grade 2 and high-risk human papillomavirus type. The authors found that increased expression of IDO1 may contribute to immune avoidance and an increased frequency of disease progression in HPV16 and 18 positive lesions.  The paper is well written. However, some issues remain.

All the acronyms must be explained at their first appearance in the text (e.g., IDO1).

This has been corrected.

Moreover, a description of IDO1 should be added in the Introduction.

Thank you for the suggestion, which improves the manuscript clarity.  The following has been added to the introduction (and removed from the discussion).

Indoleamine 2,3,-dioxygenase 1 (IDO1) is an intracellular cytosolic enzyme that mediates local suppression of T cell proliferation and activation by catalysing tryptophan to kynurenine.  Tryptophan degradation and accumulation of kynurenine in the extracellular microenvironment suppresses effector T cell proliferation and activation [10, 11].  Inhibition of IDO1 function has been shown to elevate tumour-infiltrating CD8 T cells in the mouse [12], and in vitro [13].’

In table 2, the authors should specify which third (1/3, 2/3, or 3/3) is the lower one.

Thank you.  This has now been added in the footnote to table 2.

More Kaplan-Meyer estimator curves according to immune microenvironment may be helpful.

We thank the reviewer for the suggestion and refer them to our recent publication, which includes the Kaplan-Meyer curves for the progression versus persistence/regression endpoints:  Saito, M., A. Rajesh, C. Innes, R. van der Griend, P. Fitzgerald, B. Simcock, P. Sykes, and M. Hibma, Blimp-1 is a prognostic indicator for progression of cervical intraepithelial neoplasia grade 2. J Cancer Res Clin Oncol, 2022. 148(8): p. 1991-2002.

Round 2

Reviewer 2 Report

Thank you for improving the manuscript.